# A Novel Dual of Shannon Information and Weighting Scheme

## Abstract

Shannon Information theory has achieved great success in not only communication technology where it was originally developed for but also many other science and engineering fields such as machine learning and artificial intelligence. Inspired by the famous weighting scheme TF-IDF, we discovered that Shannon information entropy actually has a natural dual. To complement the classical Shannon information entropy which measures the uncertainty we propose a novel information quantity, namely troenpy. Troenpy measures the certainty and commonness of the underlying distribution. So entropy and troenpy form an information twin. To demonstrate its usefulness, we propose a conditional troenpy based weighting scheme for document with class labels, namely positive class frequency (PCF). On a collection of public datasets we show the PCF based weighting scheme outperforms the classical TF-IDF and a popular Optimal Transport based word moving distance algorithm in a kNN setting with respectively more than 22.9% and 26.5% classification error reduction while the corresponding entropy based approach completely fails. We further developed a new odds-ratio type feature, namely Expected Class Information Bias(ECIB), which can be regarded as the expected odds ratio of the information twin across different classes. In the experiments we observe that including the new ECIB features and simple binary term features in a simple logistic regression model can further significantly improve the performance. The proposed simple new weighting scheme and ECIB features are very effective and can be computed with linear time complexity.

## 1 Introduction

The classical information theory was originally proposed by Shannon (1948) to solve the message coding problem in telecommunication. It turned out that it has far more profound impact beyond communication theory, and it has shaped all aspects of our science, engineering and social science by now. The core concept entropy was coined to measure the expected rareness or surprise of a random variable $X$ across its distribution. In the literature entropy is usually taken for granted as the *information* in many people's mind. The mutual information (MI) between two variables is the difference of the entropy of a variable from its conditional entropy given the other variable. MI maximization principle also has been studied and used widely in machine learning. Recently MI has also been employed as part of the objective function for optimization in neural network models based representation learning(Belghazi et al., 2018; Hjelm et al., 2019).

Along another line, weighting scheme has been used extensively in information retrieval tasks. Term Frequency-Inverse Document Frequency(TF-IDF), a simple statistic heuristic proposed by Sparck Jones (1972) has been widely used as a weighting method over half a century in information retrieval and natural language processing. It weighs down a term if its document frequency increases in the corpus, as it becomes less effective to distinguish from others when it gets popular and its appearance brings less *surprise* in the sense of Shannon self-information. This simple but effective algorithm has achieved great success as a robust weighting scheme. Even today many search engines and digital database systems still employ TF-IDF as an important default algorithm for ranking.

In the past decades a few researchers have intensively investigated on it for a better theoretical understanding of the underlying mechanism rather than a heuristic and intuition argument. Robertson (2004) justified it as an approximate measure of naive Bayes based probability relevance model in

information retrieval. Some researchers tried to explain from the information theory point view. Aizawa (2003) interpreted it as some probability weighted amount of information. Siegler & Witbrock (1999) interpreted IDF for a term exactly as the mutual information between a random variable representing a term sampling and a random variable representing a document sampling from a corpus. Many other variants of the term frequency have been proposed in the literature. For example, BM25(Robertson, 2009) based on probabilistic retrieval framework was further proposed and it has been widely used by search engines to estimate the relevance of documents to a given search query. In general the derived applications go far beyond text processing and information retrieval community.

The connection between TF-IDF and information theory mentioned above is quite motivating. This makes us wonder if there are other simple and effective weighting schemes that can be established from information theory. In order to achieve this goal, it turns out that we first developed a new metric of information quantity for certainty, namely *troenpy*, a natural dual to entropy, and then used it to derive a new type of weighting scheme which works very well in the extensive experiments as we hoped.

In the following we will first introduce troenpy and its basic properties, and share some insights we have for this innovation. Then for the classical task of supervised document classification, we will develop a troenpy based weighting scheme for document representation. This weighting scheme makes use of the documents class label distribution and helps improving the model performance very significantly. Employing both entropy and troenpy, we will also define some new odds-ratio based class bias features leveraging the document class label distribution. Finally evaluating under the simple KNN and logistic regression settings, we show that the proposed new weighting scheme and new features are very effective and achieved substantial error reduction compared with the TF-IDF and a popular optimal transport based document classification algorithm on a collection of widely used benchmark data sets.

## 2 DUAL OF SHANNON ENTROPY

We fix the notations first. Here we let $X$ indicate a discrete random variables with probability mass function $p_X(x)$. The Shannon entropy (sometimes also called self-information) measures the uncertainty of the underlying variable, or the level of *surprise* of an outcome in literature. To understand this, note when the event is rare, that is the probability $p_X(x)$ is small, the measure $-\log(p_X(x))$ is large; when the event is not rare, that is the probability $p_X(x)$ is not small, the measure $-\log(p_X(x))$ is not big. Therefore in this sense of Shannon, the measure $-\log(p_X(x))$ does represent the rareness or surprise degree of an event. In this work we purposely call it Negative Information(NI) for showing the duality nature later. That is,

$$\text{NI}(x) := -\log(p_X(x)) = \log\frac{1}{p_X(x)}. \tag{1}$$

Now since Shannon information measures *surprise*, can we measure the *certainty* or *commonness* instead? This is exactly the contrary to the Shannon information, the dual of Negative Information Entropy. This motivates our definition below.

**Definition 2.1.** We define Positive Information (PI) of an outcome $x$ as

$$\text{PI}(x) := -\log(1 - p_X(x)) = \log\frac{1}{1 - p_X(x)}. \tag{2}$$

To understand why PI measures the certainty of an event, note when the event is rare, that is the probability $p_X(x)$ is small and the certainty is small, the measurement $-\log(1 - p_X(x))$ is very small; when the event is not rare, that is the probability $p_X(x)$ is large and the certainty of the event is large, the measurement $-\log(1 - p_X(x))$ is also large. So the PI can measure the certainty of an event faithfully in the same sense of Shannon.

For discrete random variables with probabilities $p_i$, where $i \in \{1, \ldots, K\}$, the value PI=$\log(\frac{1}{1-p_i})$ is the measure of *non-surprise* or *commonness*. Note from the definition, PI has the same value range $[0, \infty)$ as NI. A conventional way to avoid the infinity value ranges numerically is to add a small value epsilon to the denominator, and one can choose the epsilon value according to desired resolution. Note if we denote $\bar{x}$ the complement of outcome $x$, then $\text{PI}(x) = \text{NI}(\bar{x})$.

Naturally by taking expectation across the distribution, we propose a dual quantity of entropy, namely **troenpy**, to measure the certainty of $X$. Troenpy is simply the expected *positive* information, while entropy is the expected *Negative* Information (NI). Troenpy reflects the level of *reliability* of the $X$ outcomes that the data conceals.

**Definition 2.2.** The troenpy of a discrete random variable $X$ is defined as the expectation of the PIs,

$$T(X) := -\sum_x p_X(x)\log(1 - p_X(x)). \tag{3}$$

For continuous random variable $X$ with density function $f(x)$, the differential troenpy is formally defined by first dividing the range of $X$ into bins of length $\Delta$, and the integral within each bin can be represented as $p_i = f(x_i)\Delta$ by the Mean Value Theorem for some $x_i$ in the bin, and taking the limit by letting $\Delta \to 0$ if the limit is finite. It turns out that the integral is zero.

$$\begin{aligned} T(f) :&= -\int f(x)\log(1 - f(x))dx \\ &= \sum f(x_i)\Delta\log(1 - f(x_i)) \end{aligned} \tag{4}$$

Note the following fact about troenpy can be observed. For a discrete random variable with probabilities $p_i$, where $i \in \{1, \ldots, K\}$, troenpy achieves the maximum value infinity when an event is completely certain with corresponding probability $p_i = 1$. Note this is different from entropy, whose value is bounded and ranges from zero to the maximum $\log K$.

**Theorem 2.3.** *Troenpy achieves the minimum value $\log(\frac{K}{K-1})$ when the underlying discrete distribution is uniform with each $p_i = \frac{1}{K}$ for all $i$, while entropy achieves its maximum value $\log K$.*

*Proof.* To see why troenpy achieves such minimum value, note that the sum $\sum_{i=1}^{K}(1 - p_i) = K - (p_1 + \ldots, p_K) = K - 1$. If we let $q_i = (1 - p_i)/(K - 1)$, then $q = (q_1, \ldots, q_K)$ is a probability distribution. According to the Gibbs inequality (MacKay, 2003), the cross entropy $-\sum_{i=1}^{K} p_i \log q_i$ achieves minimum value when $p_i = q_i$, which immediately gives $p_i = 1/K$. It is also obvious that the troenpy can be treated as the above cross entropy minus the constant $\log(K - 1)$. $\square$

Note conceptually we can regard troenpy as a complimentary metric of information in a distribution in the sense of reliability. It measures how much confidence about the outcomes in a distribution. If the certainty increases, it means some outcomes gain more confidence and the uncertainty of the outcomes decreases correspondingly. Because of the intrinsic nature of troenpy, it naturally serves as a weighting scheme measuring the reliability of a random variable. More certainty means more predictability. If a random variable has very low certainty, this just means it has a high entropy and is very noisy. Thus it is not a good feature for prediction purposes and should be correspondingly down-weighted.

Next we define conditional troenpy which will motivate and lead to the weighting scheme in next section. Let $p(x, y)$ denote the joint distribution of the discrete random variables $X$ and $Y$, and lowercase letters denote the random variable values.

**Definition 2.4.** We define the **Conditional Troenpy** of X given Y, denoted as $T(X|Y)$, to be the following $T(X|Y) = \sum_y p(y)T(X|Y = y)$.

It can further be reduced to the following

$$\begin{aligned} T(X|Y) &= -\sum_{y,x}[p(y)p(x|y)\log(1 - p(x|y))] \\ &= -\sum_{x,y} p(x, y)\log(1 - p(x|y)) \end{aligned}$$

**Definition 2.5.** We define the **Pure Positive Information** of $X$ from knowing $Y$, denoted as PPI$(X; Y)$, to be the troenpy gain $T(X|Y) - T(X) = \sum_{x,y} p(x, y)\log\frac{1 - p(x)}{1 - p(x|y)}$.

Note PPI$(X;Y)$ is the analogue of the classical mutual information. It measures the troenpy change due to the presence of another random variable. Thus this PPI can serve as a candidate for weighting scheme. Note in general PPI$(X;Y) \neq$ PPI$(Y;X)$. This is very different from the mutual information MI$(X;Y)$ of two random variables $X$ and $Y$ in the literature, where MI$(X;Y) =$ MI$(Y;X)$. In order for them to be equal, this requires $(1-p(x))/(1-p(x|y)) = (1-p(y))/(1-p(y|x))$, which is equivalent to $p(x) - p(x|y) = p(y) - p(y|x)$. However, this last equation does not hold in general.

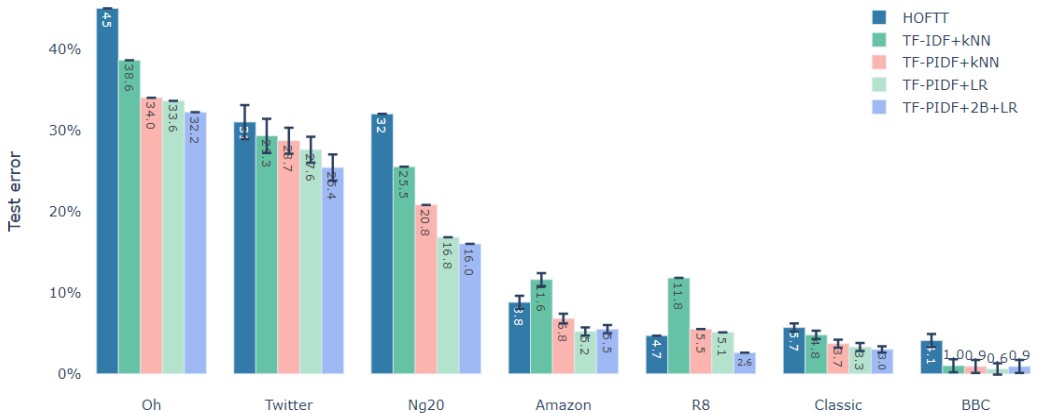

Figure 1: Errors of document classification for 7 Datasets with TF-IDF and TF-PI

## 3 WEIGHTING SCHEME FOR SUPERVISED DOCUMENTS CLASSIFICATION

In this section we first briefly review the information theoretic interpretation of TF-IDF, then naturally we define a new weighting scheme using the newly proposed troenpy as an analogue.

### 3.1 REVIEW OF IDF

Here we follow the information theoretic view mentioned above (Aizawa, 2003). We consider the classical text documents classification task in the routine supervised learning setting. The typical scenario is that given a corpus collection of documents $D_1, \ldots, D_n$, where $n$ denotes the total number of documents. Each document $D_i$ has a class label $y_i$ from a finite class label set $Y = \{1, 2, \ldots, K\}$, where $K$ is the total number of classes. For a given word $w$, let $d$ denote the number of documents where $w$ appears. Then the IDF is simply given by the following:

$$\text{IDF}(w) = 1 + \log \frac{n}{1+d} \tag{5}$$

It can be interpreted as the *self* (negative) information in information theory, which measures the *surprise* of the word $w$. The idea follows as below: Fix a word $w$ with document frequency d in a collection of $n$ documents, then the probability of $w$ appearing in a document D can be approximated by Prob$(w \in D) = \frac{d}{n}$. Then the negative-information NI$(w) = -\log\text{Prob}(w \in D) = \log\frac{n}{d}$. To smooth out the case when $d = 0$, adding 1 to the denominator gives NI$(x) \approx \log\frac{n}{d+1}$. Also, the summation of all TF-IDFs, each of which represents bits of information weighted by the probability of a word, also recovers the mutual information between words and documents.

### 3.2 POSITIVE CLASS FREQUENCY

In this section we will make use of the document class distribution and define a new term weighting method, which can be applied later for the classification task. First for all the $n$ documents in the corpus, we collect the counts of documents for each class. We denote the class label distribution as $C = \{C_1, \ldots, C_K\}$, where $C_i$ is the count of the $i^{th}$ class label. Normalizing by dividing the total number of documents $n$ gives the probability distribution $\overrightarrow{c} = \{c_1, \ldots, c_K\}$, where $c_i = \frac{C_i}{n}$.

This vector $\overrightarrow{c}$ contains the global distribution information and we can define two intrinsic quantities measuring the certainty and uncertainty.

**Definition 3.1.** We define Positive Class Frequency(PCF) for $C$ as the troenpy of $\overrightarrow{c}$. Similarly, Negative (or Inverse) Class Frequency(NCF or ICF) as the entropy of $\overrightarrow{c}$.

$$\text{PCF}(C) := Troenpy(\overrightarrow{c})$$
$$\text{NCF}(C) := Entropy(\overrightarrow{c}) \tag{6}$$

For the whole document collection (abbreviated as $\text{DC}_*$), the PCF of the normalized label vector $\overrightarrow{c}$, denoted as $PCF_*$ is a constant for each term indicating the certainty level of the whole label distribution at the collection population level. Restricting to the documents with the term $w$ present (abbreviated as $DC_1$), the corresponding conditional PCF is denoted as $\text{PCF}_1$. Similarly, $\text{PCF}_{-1}$ denotes the PCF for documents without the term $w$ (abbreviated as $DC_{-1}$). We propose using the difference $\text{PCF}_1 - \text{PCF}_*$ between $\text{PCF}_1$ and $\text{PCF}_*$ as a term weighting reflecting the certainty gain due to the presence of the term $w$. Note this is the same as the PPI introduced in last section, i.e, the conditional troenpy gain condition on the knowledge of the presence or absence of the term $w$. Without abuse of notation, we simply keep using PCF to denote this new weighting scheme. Note in the classical TF-IDF setting and general machine learning literature, such label distribution information is usually used in some supervised ways (Ghosh & Desarkar, 2018). It has not been made use of in such a simple and principled way before.

To combine the IDF and PCF weightings, we propose using their multiplication PCF·IDF, abbreviated as **PIDF**, as the weighting. Note the IDF computation uses only the term document frequency information across the corpus while the PCF leverages the documents label information via the conditional troenpy. So the simple product model make use of both corpus information about document frequencies as well as the document label information. Hence multiplying with the term frequency gives the name **TF-PIDF**. So in our setting each document can be represented as a vector of word term frequencies multiplied with selected weighting method applied such as $doc_i = [tf_1\text{PIDF}_1, \ldots, tf_m\text{PIDF}_m]$, where $tf_i$ denotes the term frequency for the $i^{th}$ token in literature and $m$ is the number of unique selected terms in a document.

On the other hand, the entropy based weighting NCF · IDF is correspondingly abbreviated as **NIDF** and multiplying the term frequency gives **TF-NIDF**. Note the NCF is not suitable for weighting as they are the negative information measuring the uncertainty. The rationale behind this is that when a mathematical model predicts things, it relies on the learned certainty from the data, not the uncertainty. This intrinsic nature of certainty determines troenpy is the right candidate. To support this view, we will illustrate **TF-NIDF** is ineffective in the experiment session.

Next we will give another two measures which are easy for researchers to come up as candidates for the quantification of certainty. To clear the curiosity and doubts from readers, we include these measures in the experiment for performance comparison.

**Definition 3.2.** We define Reciprocal Negative Class Frequency(RNCF) for $C$ to be the reciprocal of NCF(C).

$$\text{RNCF}(C) := \frac{1}{\text{NCF(C)}} \tag{7}$$

The corresponding weighting RNCF · IDF is denoted as **RNIDF** and multiplying term frequency gives **TF-RNIDF**. The motivation for this definition is that intuitively some people may think that the RNCF reverses the changing direction of entropy, so it can be used to denote the certainty.

Instead of taking the reciprocal of entropy, another similar idea that is easy to come up is by subtracting the entropy from the maximum possible entropy of the underlying distribution. When the random variable $X$ is discrete with $K$ total number of classes, the maximum possible entropy value equals $\log K$; when $X$ is a continuous random variable, the maximum possible entropy value equals the corresponding Gaussian entropy with the same mean and standard variance (Thomas & Joy, 2006). This measure **Negentropy** first appeared in a book by Erwin (1944), and then it was further studied by Leon (1953).

**Definition 3.3.** We define Negative Entropy Class Frequency(NEGECF) for $C$ to be the difference between the maximum entropy of $C$ and NCF(C).

$$\text{NEGECF}(C) := \log K - \text{NCF(C)} \tag{8}$$

The corresponding weighting NEGECF·IDF is denoted as **NEGEIDF** and multiplying term frequency gives **TF-NEGEIDF**.

Negentropy measures the remaining uncertainty for the underlying distribution. Similarly as the reciprocal of entropy, it also reverses the changing direction of entropy. So both RNCF and NEGECF qualitatively align well with the direction of certainty in general. One problem of using the Negentropy as the measure for certainty is that when the discrete distribution $C$ is evenly distributed across all $K$ classes, the Negentropy gives zero while each class still has $\frac{1}{K}$ probability and the corresponding certainty value should be nonzero though it could be small. In this case troenpy gives $\log\frac{K}{K-1}$. So the Negentropy is essentially a measure of deviation from the even distribution for discrete random variables by definition. One can also define a measure by subtracting the entropy from other selected constants. For example, Wang et al. (2021) defined a similar weighting measure by setting the constant to be 1 and the entropy log base to be the number of classes, so the measure lies in $[0, 1]$. However, these types of measures have the same issue.

Numerically the amount of certainty measured by troenpy cannot be expressed as an analytic function of entropy. We will derive the transcendental relationship between entropy and troenpy elsewhere. In the experiment session we will show that all NCF, RNCF and BEGECF are not effective as weighting schemes. This justifies the necessity of introducing the troenpy as a corresponding measure of certainty.

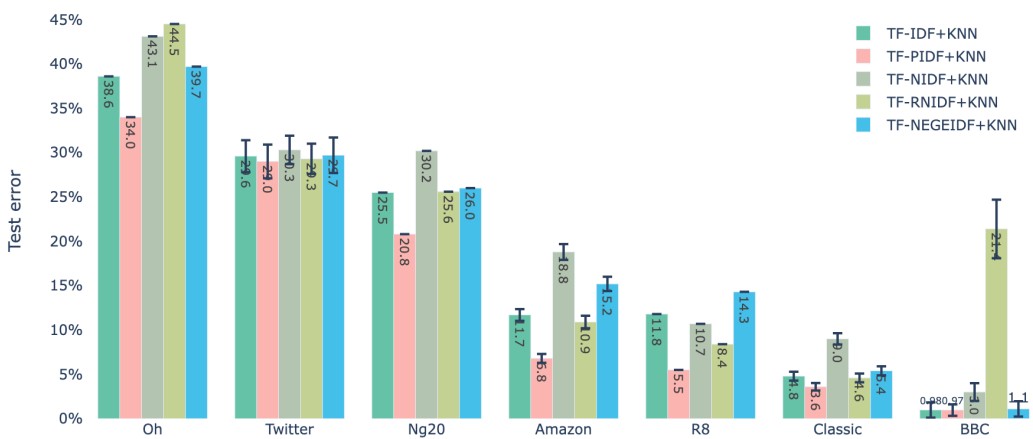

Figure 2: Errors rates of TF-IDF, TF-PIDF, TF-NIDF, TF-RNIDF and TF-NEGEIDF across seven datasets in a KNN setting

# 4 CLASS INFORMATION BIAS FEATURES AND BINARY TERM FREQUENCY FEATURES

In this section we introduce two types of features for document representation: the odds ratio based features for class information distribution and a simple binary term frequency feature. For brevity, we denote these two features as 2B features in the experiments.

## 4.1 ODDS-RATIO BASED CLASS INFORMATION BIAS FEATURES

The idea is that both the TF-IDF and TF-PIDF are obtained from a term frequency multiplied with a weight information quantity measuring their rareness or certainty, instead we can weight these term frequencies by how biased they are distributed across the classes. This idea was inspired by an algorithm called Delta-IDF. In a simple two class sentiment classification setting, Martineau & Fanin (2009) proposed first taking the difference of the IDFs between the documents of the positive class and the documents of the negative class and then multiplying with the term frequency to give their delta-TFIDF. That is, $tf_w[\log\frac{P}{P_w} - \log\frac{N}{N_w}]$, where $P$ and $N$ respectively stand for the total numbers of positive documents and negative documents, and the $P_w$ and $N_w$ respectively stand for

the total numbers of positive documents with the term $w$ appearing and the total number of negative documents with term $w$ appearing. So the difference between the IDFs of the two collections of documents are exactly the odds ratio of the documents counts for the two complementary collections of documents, which can be rewritten as $\log \frac{PN_w}{P_w N}$.

Motivated by the above, we can first compute the NCF and PCF difference for any class $i$, which gives the Class Information Bias (CIB) features. And then we take the weighted average of such CIB features across all $K$ classes. We call these new features the Expected Class Information Bias (ECIB) features. Specifically for a term $w$, we first use $n_w$ to denote the number of documents with $w$ present and $n_{iw}$ to denote the number of documents with class label $i$ and w present. Then the NCF based CIB for class $i$ is given as $\text{CIB}_i(w) = \log \frac{C_i}{1+n_{iw}} - \log \frac{n-C_i}{1+n_w-n_{iw}}$, as $(n-C_i)$ stands for the total documents not in class $i$ and $(n_w - n_{iw})$ stands for the total number of documents not in class $i$ but with $w$ appears. Similarly, the PCF based CIB is given as $\log \frac{C_i}{1+C_i-n_{iw}} - \log \frac{n-C_i}{1+n-C_i-n_w+n_{iw}}$.

Therefore for each term $w$, we can define two such distributed Class Information Bias features, one using NCF and one using PCF. The expected CIB features are precisely given by the following.

$$
\begin{aligned}
&\text{CIB-NCF}(w): \\
&= \sum_{i=1}^{K} \frac{C_i}{n} (\log \frac{C_i}{1+n_{iw}} - \log \frac{n-C_i}{1+n_w-n_{iw}}) \\
&\text{CIB-PCF}(w) := \sum_{i=1}^{K} \frac{C_i}{n} (\log \frac{C_i}{1+C_i-n_{iw}} \\
&- \log \frac{n-C_i}{1+n-C_i-n_w+n_{iw}})
\end{aligned}
\tag{9}
$$

The effect for this ECIB feature is that words that are evenly distributed for their contribution of the information quantities in a class and the rest of the class get little weight, while words that are prominent in some class for their contribution of the information quantities get more weight. So the terms characterizing specific classes are relatively better weighted as they are more representative.

### 4.2 BINARY TERM FREQUENCY

The binary term frequency (BTF) is simply a binary feature for each term $w$. $\text{BTF}(w)$ is 1 if $w$ is present in a document and it is 0 if it is absent in a documentJain et al. (2020). BTF gives the most naive representation of a document, regardless of frequency counts. We notice that BTF features are actually quite informative and together with TF-IDF can significantly improve the classification performance in the kNN setting. One can achieve this by simply summing the TF-IDF based pairwise document distance and the BTF features based document pairwise distance as the final document pairwise distance.

## 5 DATASETS AND EXPERIMENT

The goal of our experiments in this section is to validate our proposed weighting schemes and features for the supervised document classification tasks, and compare with the baseline algorithms. To achieve this we include seven text document datasets that are often used for the document classification tasks in the literature. Three datasets already have a training dataset and a test dataset split while the rest four have no such splits. The experiments of supervised document classification tasks have two settings for the evaluation: a simple kNN setting and a logistic regression setting. The evaluation metric is the error rates on the test datasets.

### 5.1 DATASETS

Here we follow closely the setup of Yurochkin et al. (2019). We use the popular seven open source datasets below for the study on kNN based classification tasks. Note these datasets have been

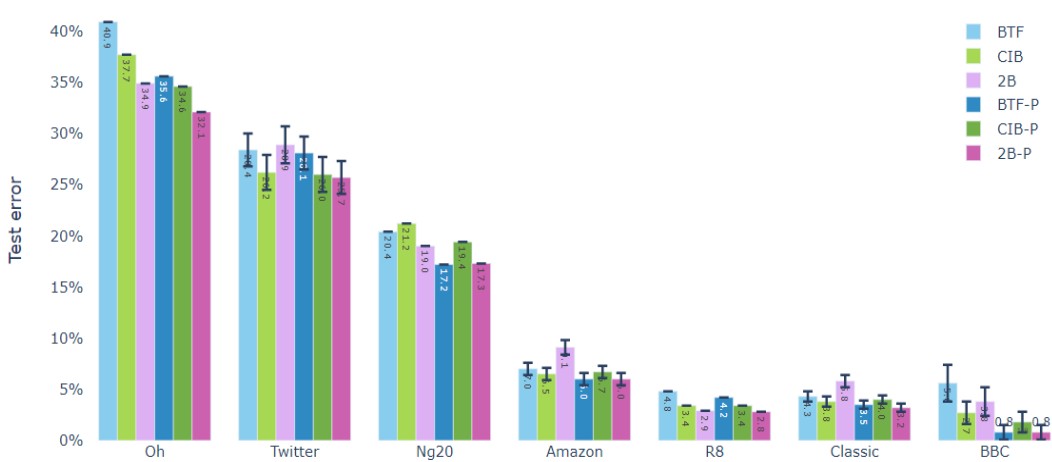

Figure 3: Error rates of document classification using 2B features in logistic regression classifier

extensively used numerous times for the classification tasks. The datasets include BBC sports news articles labeled into five sports categories (BBCsports); medical documents labeled into 10 classes of cardiovascular disease types( Ohsumed); Amazon reviews labeled by three categories of Positive, Neutral and Negative (Amazon); tweets labeled by sentiment categories (Twitter); newsgroup articles labeled into 20 categories (20 News group); sentences from science articles labeled by different publishers ( Classic) and Reuters news articles labeled by eight different topics (R8). The more detailed information about the datasets can be found in the references mentioned above. For the datasets with no explicit train and test splits, we use the common 80/20 train-test split and report the performance result based on fifty repeats of random sampling.

To minimize the datasets version mismatch, in all the experiments we use the raw text documents rather than some pre-processed intermediate formats such as some of the processed datasets provided in Kusner et al. (2015).

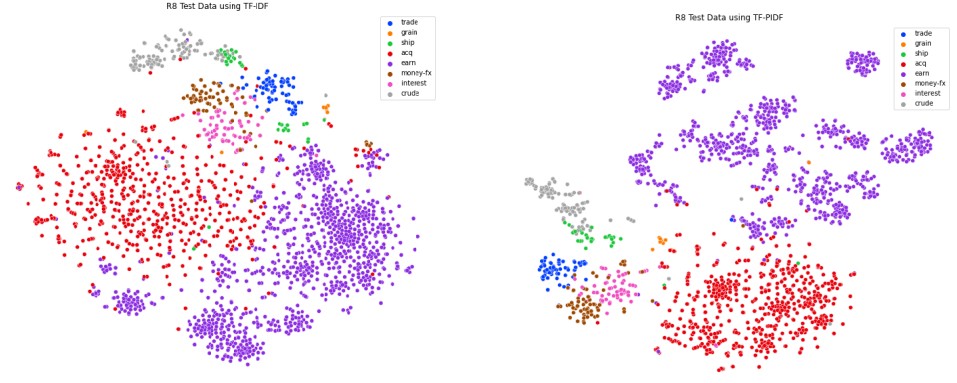

Figure 4: t-SNE on R8 data

## 5.2 EXPERIMENT SETTINGS

Here we introduce the baseline algorithms and their settings in the experiments.

**Baselines:** For the evaluation of supervised documents classification on term frequencies and their weighting, we include the classical TF-IDF document representation as a baseline. The pairwise document distance in kNN setting is computed using the TF-IDF represented vectors. For comparison purpose and reference, in the experiments we also include the result of a Word Moving Distance (WMD) based algorithm, namely HOFTT proposed by Yurochkin et al. (2019). It is a hierarchical

optimal transport distance in the topic spaces of documents. We follow closely the experiment setting of HOFTT.

**kNN based Classification:**   The features include term frequencies only. The goal is to validate the TF-PIDF weighting and compare with TF-IDF. The data pre-processing starts with removing the frequent English words in the stop word list, which can be found in the above references. To ease the kNN evaluation part, we fix the number of closest neighborhoods $k=7$ rather than dynamically selecting the optimal $k$. We compute the integrated weighting PIDF as the product of PCF and IDF, and compare with the IDF weighting for each term frequency. Using the TF-PIDF and TF-IDF, we obtain the bag-of-words vector representation of each document and take their L2 normalization, and then compute the document pairwise distance following the standard kNN procedures. Again our main goal here is to assess if the proposed PCF weighting is effective and can help improve the classical TF-IDF method. Also we want to evaluate the entropy based approach and see if it fails as we expected.

**Logistic Regression based Classification:**   In this setting we simply replace the simple kNN with a standard logistic regression model instead. In the experiments we use the Sklearn package implementation with default settings. Here we have two goals to evaluate. First we need to evaluate if the models have performance improvement when the 2B features are included, compared with the models using only the TF-PIDF features. So we can assess if 2B features are effective for the document classification task. Second we want to evaluate the PCF weighting effect on the ECIB and BTF features both separately and jointly.

Here the data preprocessing is identical to the kNN classification settings above. We mainly consider three types of features in the experiment, namely the TF-PIDF features, binary term features (BTF) and the ECIB features.

# 6   RESULTS

**kNN based Classification Experiments:**   In Figure 1 we can visually observe that the TF-PIDF based kNN model uniformly outperformed the classical TF-IDF based kNN across all seven datasets and the improvement is quite substantial for most cases with an average overall error reduction 22.9%. Noticeably the R8 dataset achieves the most 53.4% error reduction. Compared with HOFFT, the TF-PIDF achieves even more error reduction with the average of 26.5%. These uniform improvement can be explained as the PCF weighting does effectively leverage the certainty and common similarity of class label distributions across the corpus at a term level. For a term, the more PCF it has the better prediction capacity it has. For example, the different news groups in Ng20 actually share many non-stop words in common and some groups are very relevant. The learned similarity information about one group is helpful at predicting a relevant group. We also observe only slight improvement on the Twitter and BBC sport datasets which might be simply due to the small sample sizes. The Twitter has 3115 samples and BBCsport has only 737 samples, which are quite small compared with other datasets. Additionally, the Twitter sentiment dataset has three class labels consisting of positive, neutral and negative. The extreme polarity of the classes is often consistent with the fact that relatively less common description words are shared across the classes.

**PCF, NCF, RNCF and NEGECF Comparison:**   To compare the performance of TF-PIDF, TF-NIDF, TF-RNIDF and TF-NEGEIDF with the baseline TF-IDF, we did another experiment, where the datasets with no given train/test splits are resampled fifty times. The result is reported in Figure 2. We observed that the TF-PIDF is consistently effective on reducing the errors compared with TF-IDF while all TF-NIDF, TF-RNIDF and TF-NEGEIDF fail in reducing the errors as expected. Note the TF-RNIDF performs badly on the BBC dataset, which suggests that RNCF could be very unstable for some datasets. This clearly justifies that our proposal of Troenpy is necessary and troenpy cannot be substituted by entropy or its reciprocal or difference from a constant for the weighting role.

**t-SNE:**   We also use the popular t-SNE by van der Maaten & Hinton (2008) to visualize the TF-IDF and TF-PIDF classification effect on the R8 dataset. In Figure 4, the TF-PIDF appears to cluster relatively closer for each class labels and clusters are relatively separated from other cluster groups.

**Word Moving Distance Methods:** In the experiments a modern Optimal Transport (OT) based Word Mover's Distance (WMD) approach HOFTT performs poorly compared with the TF-PIDF weighting on all dataset except on R8 dataset, on which it is also outperformed by TF-PIDF employing the additional 2B features. However we are also aware of another advanced WMD method Wasserstein-Fisher-Rao(WFR) developed by Wang et al. (2020), which uses the Fisher-Rao metric for the unbalanced optimal transport problem. The reported result of WFR is comparable to our proposed methods across the datasets. Unfortunately there are some version mismatch for some datasets as well as slightly different sampling procedure for datasets with no pre-specified train-test splits, so we did not include the corresponding result in our figures. Note also that the general Sinkhorn based algorithms for such OT optimization problems have relatively high computational complexity and so they are quite expensive on computational cost. While the proposed weighting scheme and ECIB features can be obtained in a single scan of the data and the time complexity is linear, they are fast and a lot cheaper on computational cost.

**Logistic Regression based Experiments:** In Figure 1 we observed the following: (1) for the same TF-PIDF feature set, the logistic regression model uniformly outperforms the kNN approach across all datasets. This is not surprised as the logistic regression optimizes the term coefficients for optimal fitting the data while the kNN is rigid as given. (2) adding the 2B features of binary term frequency (BTF) and expected class information bias (ECIB) further significantly reduces the errors on most datasets. Compared with TF-IDF, the average error reduction is $35\%$. Compared with HOFFT, the error reduction reaches $43.4\%$. For the BBC dataset we observed a relatively large error increase, and we hypothesize that this may be due to the very small test sample size of the dataset.

In Figure 3 we reported the results of using BTF and ECIB features in the logistic regression setting. We observed the following. Both BTF and ECIB features are quite effective when used individually alone. ECIB performs better than BTF on all datasets except on the dataset of 20 Newsgroup, where they are relatively close. Simply combining the two features together not necessarily always improves the performance, instead it leads to slightly more errors on a couple of the datasets. We also observe that applying the PCF weighting helps on majority of the cases. Visually the left three bars of light color represent 2B features without PCF weighting while the right three bars of darker color represent corresponding features with PCF weighting applied.

# 7 DISCUSSION

The current work first proposed a new information measurement of certainty and an associated weighting scheme leveraging the document label information, and further demonstrated its effectiveness on several popular benchmark datasets of English text documents. We also gave a couple of measures which are easy to come up as candidates for certainty. Unfortunately neither of them is effective as a weighting scheme. Our troenpy is mathematically the canonical dual of entropy from the definition. For documents without label information available, the current proposal cannot apply directly. However, a few unsupervised tasks often can be reformulated into popular self-supervised problems. The only difference from the above supervised setting is that the labels and features are from the same space, and we can apply the developed methods to process without much difference. In modern NLP community distributed representations of word tokens are widely used in language models for various tasks. The proposed troenpy and the weighting schemes can actually be integrated into neural network based large language models and can further lower the perplexities. For computer vision, the corresponding methods can be suitably modified and then applied to vision language models for downstream tasks. We will leave these as future work.

# 8 REPRODUCIBILITY STATEMENT

Data and codes for the experiments can be found in the supplemental material and all results can be easily reproduced.

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
