# OpenReview forum: "A Novel Dual of Shannon Information and Weighting Scheme"
_ICLR.cc/2025/Conference — Submitted to ICLR 2025_

### Official Review · Reviewer_5FaU · 2024-10-28

**Soundness:** 3
**Presentation:** 3
**Contribution:** 3
**Rating:** 6
**Confidence:** 5

**Summary:**

Note added on December 3: I am glad that the authors found my review to be useful. I agree with them that the other review with a score of 1 is too extreme, in its score and its wording. However, the authors need to understand better the limitations of their submission, and what is required by a selective venue such as ICLR.

"We did have other successful applications such as the large language models, recommendation system and computer vision etc, which we treated elsewhere as a series of following work" is not a reason to accept the current paper. Acceptance should be based only on actual contributions, not on unsubstantiated claims or promises.

The experimental section of the submission is quite weak, as pointed out previously. The authors did not cite, and did not compare to, the state-of-the-art in the family of TFIDF-based methods for document classification. The BNS method is just one of these methods. It is the duty of the authors to do a careful literature search, and to convince reviewers that the submission includes fair comparisons with the previous best related methods. The submission does not succeed currently in this direction.

=============

This paper introduces a variation of entropy that can be called entropy of the complement probabilities, $-\sum_i p_i \log(1 - p_i)$. The paper uses this as a weighting approach for words in document classification, and gets better accuracy than using alternative weighting methods.

**Strengths:**

Entropy of the complement probabilities has not been studied before in the research literature, and makes sense as a useful measure. The experimental results are believable.

**Weaknesses:**

Section 3 and the experiments based on it are clear and persuasive. Section 4 is less clear.

The experimental work is actually very limited, using only seven small datasets and essentially copying the experimental design of just one previous paper.

Although this submission is interesting, it is about document classification based on bag of words, which is mostly obsolete given the availability of large language models which understand document semantics, and hence can classify text with much better accuracy.

**Questions:**

015: Troenpy is not a "dual" in a precise mathematical sense. It is an interesting and novel variant.

053: This is a very incomplete summary of theoretical justifications for TFIDF. See also, among others, https://link.springer.com/chapter/10.1007/11575832_33, Deriving TF-IDF as a Fisher Kernel.

072: Cite and compare experimentally to previous methods that use odds ratios. See https://dl.acm.org/doi/abs/10.1145/1458082.1458119 "BNS feature scaling: an improved representation over tf-idf for svm text classification" and https://www.jmlr.org/papers/volume3/forman03a/forman03a_full.pdf "An Extensive Empirical Study of Feature Selection Metrics for Text Classification.

114: The new concept, troenpy, is similar to the so-called "one versus all" approach to multiclass classification (which more precisely should be called "one versus rest").

119: "It turns out that the integral is zero." This sentence seems out of context and unsupported. The whole paragraph 116 to 124 seems incomplete and not needed.

203: It use confusing to use the word "frequency" for a value such as d that is an integer count. Use the word "count" as on line 214.

285: The word "negentropy" may appear in the 1944 book by Schrodinger, but presumably not as a formal measure, since the book predates Shannon. Note that the reference is formatted incorrectly, since Schrödinger is the author's family name, not Erwin.

357: Binary term frequency is a standard approach for document classification. See http://kamalnigam.com/papers/multinomial-aaaiws98.pdf, "A Comparison of Event Models for Naive Bayes Text Classification."

403: Does the absence of a validation set mean that there is absolutely no hyperparameter optimization?

429: It seems that all methods use raw TF, i.e., raw word counts. It is well-known that log(TF) or some other squashing function of TF usually gives better accuracy. The reason is burstiness, as discussed in the paper mentioned above, "Deriving TF-IDF as a Fisher Kernel."

490: It is not good enough to just say that previous work used different datasets. Get or implement Wang's method and run it on your datasets.

502: The meaning of "adding" is not clear.

506: Do additional experiments to explain the error increase; do not merely speculate.

**Details Of Ethics Concerns:**

None.

---

> ### Author Response · Authors · 2024-12-03
> **The authors thank the reviewer for the detailed questions and advice improving the article!**
>
> The authors first thank the reviewer 5FaU for the numerous detailed and constructive comments! The document classification task is quite old indeed. There are three reasons that we still use it here: (1) the original idea of troenpy came from the investigation of TFIDF for document classification; (2) it is a simple task for everyone to evaluate the proposal.; (3) a slight variation of the weighting scheme has been successfully applied to the popular language models with very significant perplexity reduction. We did have other successful applications such as the large language models, recommendation system and computer vision etc, which we treated elsewhere as a series of following work. If it is an effective measure for certainty, it is natural for people to validate on simple tasks before trying more complex tasks.
>
> 015: The “Duality” here means certainty vs uncertainty, not mathematically. But both entropy and troenpy are statistical expectations of a distribution.
>
> 053: The authors would include the suggested article “Deriving TFIDF as a Fisher Kernel” in the reference.
>
> 072: The authors would cite and include the BNS method in the experiment comparison. It is an interesting work.
>
> 114: The reviewer is right. Exactly as pointed out, the probability (1-p) refers to all the rest of scenarios, i.e, the complement probability.
>
> 119: Here the continuous version is included simply for the completeness of troenpy.
>
> 203: The authors would follow the advice and use count instead of frequency.
>
> 285: The authors also realized the negentropy formatting issue and would update the name.
>
> 357: The authors would include the suggested references on Binary term frequency.
>
> 403: Yes. In the experiments there is no hyperparameter tuning and every hyperparameter is fixed.
>
> 429: Thanks a lot for pointing out the log(TF)! The TFs used in the experiments are the raw counts divided by the total token counts in a document, so they are not the raw counts. The paper mainly tests if the weighting is effective or not, as long as the same baseline TF is used in the methods.
>
> 490: Wang’s method is numerically equivalent to the Negentropy approach as it equals multiplying each weighting term by the log2(K). The authors would add a comment and point this out  in the text.
>
> 502: “adding” means “including”, the authors would update it.
>
> 506: the authors would do an experiment by sampling less data from a couple of other datasets and compare the result. It would validate the hypothesis if higher error rates are observed in sampled datasets.

---

### Official Review · Reviewer_fgCx · 2024-11-05

**Soundness:** 3
**Presentation:** 2
**Contribution:** 2
**Rating:** 5
**Confidence:** 2

**Summary:**

The paper presents a new metric called "troenpy," which is designed to complement Shannon entropy. While entropy measures uncertainty, troenpy aims to quantify certainty. Additionnally, the authors suggest a new weighting method for documents with class labels, called positive class frequency (PCF). They demonstrate that this method significantly outperforms other existing methods.

**Strengths:**

1- The paper is well written and quite easy to follow

2- The experimental seems to show the efficiency of the proposed method.

**Weaknesses:**

1.The paper currently lacks a rigorous theoretical foundation justifying the choice of log(1−p(x)) as the basis for troenpy. While some intuitive motivations are provided, they don’t fully explain why this particular transformation should be optimal or preferable over other possible functions, such as log(g(p(x))). To strengthen the paper, it would be helpful if you could provide a more formal theoretical analysis or justification for the choice of log(1−p(x)) over other alternatives. This could involve deriving troenpy from first principles in a way that demonstrates its uniqueness or optimality for measuring certainty. Such an approach would make troenpy more compelling by showing that this transformation is not only intuitively sound but theoretically motivated as well.

2.The paper’s methodological presentation is somewhat unclear in terms of focus. While troenpy is presented as the main contribution, it is ultimately used only as part of the Positive Class Frequency (PCF) weighting scheme, rather than being explored on its own. At the same time, other features, such as Class Information Bias and Binary Term Frequency, are introduced without a clear link to troenpy, which can dilute the focus of the contribution. To enhance clarity, it may help to more clearly separate the different contributions or to highlight how they relate to the central concept of troenpy. If troenpy is indeed the primary contribution, consider focusing on its properties and potential applications more directly, perhaps by exploring it in different contexts out of document classification or directly comparing it with other certainty measures. Alternatively, if the emphasis is on the PCF and the broader feature set for document classification, reframing troenpy as one component of a larger methodological toolkit could provide a more cmprehensive narrative.

**Questions:**

1- Given that entropy has broad applications across various fields, could you clarify whether troenpy has potential uses beyond document classification? The current scope is quite narrow, and identifying additional applications could help to better demonstrate its value and versatility.

2-  Does troenpy come with any theoretical guarantees that it is the optimal measure for assessing certainty? Could you clarify whether alternative formulations, such as using the logarithm of another decreasing function of p(x), might also be valid? Providing insights into the theoretical underpinnings of troenpy’s formulation would enhance its credibility.

3- Is the primary goal of your research to derive a new measure (troenpy) or to improve upon TF-IDF? If the intention is to present troenpy as a novel measure, it would be helpful to discuss potential applications beyond document classification, as this focus currently feels quite limited. Conversely, if the emphasis is on document classification, the paper leans toward being empirical in nature. It appears there is a mix of these two focuses, which makes the methodological framework somewhat unclear. Could you clarify your primary aim and the intended contributions of the paper?

---

> ### Author Response · Authors · 2024-12-03
> **The authors thank the reviewer for the great questions!**
>
> Lines 524-530 briefly described other applications of troenpy such as the popular language models and unsupervised learning. More applications on fields such as image and genomics etc will be given in a series of works following the current paper. In general the certainty measurement is a general and useful quantity to employ in prediction ML/AI tasks. Either of the new applications of troenpy requires more room and cannot fit the 10 page limit. The authors would rather give a deep and focused study on the document classification here.
>
>  The paper simply proposes a new measure of certainty and validates its effectiveness by working out a classical document classification problem as empirical evidence. The given weighting scheme method actually applies directly to the popular language model study with slight modification. So it is an important case study. The authors tried to keep a balance between methodology development and real data study. The troenpy was actually discovered during the weighting scheme exploration. However, it is obvious that the troenpy itself has more potential value other than the weighting scheme as it is  a fundamental measure of certainty.
>
>
>
> The question on the optimality of troenpy is a fantastic and deep question! One can start with the following four basic conditions: (1) T(p_1,..,p_n)(denoted as T(n)) is continuous in P_i’s; (2) if all p_i=1/n, then T is a monotone decreasing function of n. And T goes to 0 as n goes to infinity. When all p_i are zero except one with value 1, T is infinity. (3) when n=2, p_1=p_2=½, then T(2)=H(2)=log2. This is the only special scenario that T=H. (4) T(tp_1,(1-t)p_1,p-2,..,p_n)=T(p_2,..,p_n)+T(tp_1,(1-t)p_1). Here T(p_2,..p_n) is the generalized notion by Renyi, where the sum of p_i's is less or equal than 1. This last condition explains the certainty computation starts from the observed leaf events and then goes back to the earlier branch events. Note this direction reverses the corresponding entropy condition. One can first consider the special case n=2^m power, and argue that the T(n)=-log(1-1/2^m)). Then one can argue the general integer case using the sandwich method. Though the paper is an empirical study, the authors will make a mark outlining the formula derivation. A complete detailed proof will be given in the appendix if space permits or other following works.

---

### Official Review · Reviewer_UsGo · 2024-11-13

**Soundness:** 1
**Presentation:** 2
**Contribution:** 1
**Rating:** 1
**Confidence:** 5

**Summary:**

This paper introduces the so-called  "troenpy" which authors call dual. But I doubt whether this is a proper terminology. This measure is applied to a weighting scheme for supervised document classification.  Simple mathematical properties are presented as Theorems without having any theoretical results. Apart from this, the paper only makes negligible contributions and completely ignores the fundamentals of information theory. Experiments are very limited, and events will not be enough for undergraduate assignments. Most of the paper is filled with trivial calculations.

**Strengths:**

Apart from the enthusiasm of the authors to conceive the idea and write this paper, I do not see any major strengths in this paper.

**Weaknesses:**

This paper completely ignores the various results in information theory, coding theory, and machine learning, where Shannon entropy plays an important role (see my questions below). Apart from some trivial calculations, the paper has no theoretical contributions. The motivations or mathematical significance of so-called measures of certainty are not very apparent. There were some classical generalizations of entropy as Renyi entropy exists in the literature, and though authors call this some kind of dual, no connections to existing literature were made. One can still appreciate the results without any mathematical backing, but this paper does not show any extensive practical applications--that too considering that Shannon entropy appears in so many applications from maximum entropy methods to reinforcement learning. Please see my question below.

**Questions:**

(1) Shannon entropy acts as a lower bound for the average code length for source entropy. What is the significance of "troenpy" here?
(2) Can you show how "trophy" has significance in asymptotic equipartition property? One can show that given a discrete-time stationary ergodic stochastic process X, - 1/n log(X1, X2,...,Xn) converges to H(X), almost surely. Can you establish such results with"troenpy"?
(3) What is the so-called "dual" of Kullback Leibler divergence? Will that divergence have any role as a "distance measure" of probability measures, and what happens to Pinsker Inequality?
(4) Maximum entropy plays an important role in machine learning and image reconstruction. Where does "troenpy" fit in here?  Does the maximum entropy distributions become "minimum troenpy distributions"?
(5) Does Shannon entropy play an important role in the characterization of typical sets? Where does "troenpy" fit in here?

---

> ### Author Response · Authors · 2024-12-03
>
> (1)The reviewer UsGo did not point out any scientific flaws of the proposal and even ignored the superior performance (E.g. 20+% error reduction) on seven datasets, but claimed “the paper only makes negligible contributions ”, this is ridiculous and unacceptable!
>
> (2) UsGo did not appreciate the novel idea of measuring Certainty, a new type of information beyond Shannon. Instead UsGo described the work as a trivial and negligible contribution. It’s obvious UsGo did not understand the value and importance of the innovative work at all. Troenpy opens a new territory for studying certainty which complements the current information theory of only uncertainty and provides opportunities for various applications in many scientific fields.
>
> (3) UsGo complained the work is too simple, but it is very effective! Great truths are simple.
> As Einstein said, “Everything should be made as simple as possible, but no simpler.” The authors regard this as the best kind of praise. Note the work is mathematically as simple as Shannon entropy.
>
> (4) The questions asked by UsGo are indeed natural questions following the new concept. These counterparts of Shanon entropy will be given in a series of following papers exploring the basic charactering properties as well as new applications. This paper is just the first conference article reporting the new measurement and demonstrating its utility in a classical example that everyone can easily verify the correctness of the proposal. The paper is intentionally self-contained with the minimum theoretical argument so the machine learning practitioners can easily apply the new measurement in real word problems. Once the new concept is given, it is relatively straightforward for one who’s familiar with the basic properties of entropy in information theory to develop the corresponding properties for troenpy.  Keep in mind that the troenpy is a convex function  of the probabilities p_i’s,  it takes the minimal value  when all p_i’s are equal, that is exactly when the entropy takes the maximum value. However, the counterpart properties of troenpy are a bit more complex than that of entropy, whose  properties and relations are quite neat. This is due to the essential difference of the two reciprocal information types. The critical and difficult part is to have the insight and innovative mind and discover the new concept from new perspectives.  It is unrealistic to develop all the following work in such a short 10 page conference article.
>
> (5) The Renyi entropy is conceptually a generalization of Shannon entropy measurement, which still satisfies the critical entropy decomposition condition, see Renyi 1961 paper for details. The troenpy measures a completely different new type of certainty information from entropy, which does not meet the condition.

---

### Meta-Review · Area_Chair_LYAz · 2024-12-23

**Metareview:**

This paper considers a dual of Shannon entropy. Unfortunately reviewers have major concerns with the contributions; please revise accordingly and I encourage a stronger resubmission as there are some good ideas in this paper. While reviews were mixed, there is a lot of context from information theory missing in the presentation here

**Additional Comments On Reviewer Discussion:**

Reviews were mixed; the negative review was quite helpful in providing concrete details on what was lacking

---

### Decision · Program_Chairs · 2025-01-22

Reject